# Dynamic Cross-Layer Prefix Alignment for Resolving Label Preference Discrepancies in LLMs Fine-Tuning

## Abstract

Fine-tuning large language models (LLMs) to adapt them for specialized downstream tasks is a common practice, yet existing methods overlook a critical issue: label preference discrepancies among different annotators. Such inconsistencies in labeling can significantly impair the model's robustness and generalization. In this work, we propose Dynamic Cross-Layer Preference Correction (DCPC), a novel self-supervised learning framework designed to mitigate these inconsistencies. DCPC incorporates a preference-sensitive similarity mechanism, cross-layer prefix alignment, and a Preference Correction Module (PCM) to dynamically adjust embeddings across transformer layers. By leveraging self-supervision, DCPC effectively aligns semantic representations and ensures consistency in label predictions, even in the presence of preference shifts. We evaluate DCPC across multiple tasks using prominent base models and introduce modified datasets that simulate real-world preference shifts. Our results show that DCPC consistently outperforms state-of-the-art Parameter-Efficient Fine-Tuning (PEFT) methods in handling label preference discrepancies.

## 1 Introduction

The rapid advancement of large language models (LLMs) has not only revolutionized the field of natural language processing (NLP) but has also significantly impacted a wide range of other domains, including healthcare, finance, and education. Models such as GPT-4(Achiam et al., 2023) have demonstrated remarkable capabilities in tasks ranging from text generation(Li et al., 2024) and comprehension(Cheng et al., 2023) to complex reasoning(Wu et al., 2024). These advancements are primarily driven by large-scale pre-training on vast datasets, which allow LLMs to generalize across diverse tasks and domains. More and more downstream applications cannot afford the high costs of pre-training large models or full parameter fine-tuning(Han et al., 2021). As a result, an increasing number of Parameter-efficient Fine-tuning (PEFT) techniques have been proposed, such as LoRA(Devalal & Karthikeyan, 2018) and P-Tuning v2(Liu et al., 2021).

Despite the success of PEFT techniques, a critical issue remains largely unaddressed: **the impact of inconsistent labeling preferences across fine-tuning datasets**. Fine-tuning data often comes from various sources, with differing annotation styles and formats, which results in significant variations in the structure and consistency of the data(Mieleszczenko-Kowszewicz et al., 2023). Traditional methods like Prefix-Tuning(Li & Liang, 2021)

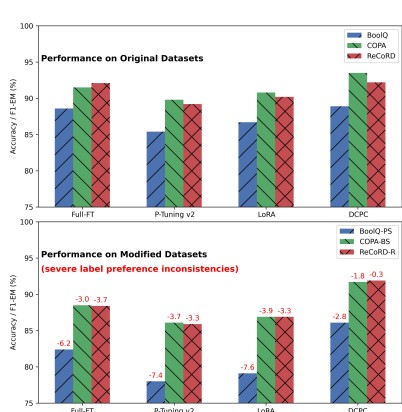

Figure 1: The performance of Full-FT, baseline PEFT Methods, and our DCPC across different datasets. The performance drop on the modified datasets, where label preference inconsistencies were introduced. DCPC exhibits significantly smaller performance drops compared to other methods.

are not designed to adapt to these structural changes, making them ill-suited for handling datasets with heterogeneous annotation practices(As is shown in Figure 1). For instance, in crowdsourced datasets, where data is labeled by workers with varying backgrounds and preferences, the inconsistency in labeling can lead to poor label agreement(Checco et al., 2017). This problem is particularly common in cases where datasets are annotated by multiple groups, such as when training data is collected from annotators with different educational levels or expertise. Such variations introduce discrepancies in the labels, making it challenging for models to generalize well.

To address the challenge of label preference inconsistencies, we propose Dynamic Cross-Layer Preference Correction (DCPC), a novel self-supervised framework specifically designed to mitigate the impact of inconsistent annotations across fine-tuning datasets. Unlike traditional fine-tuning methods(Devalal & Karthikeyan, 2018; Liu et al., 2021) that treat label discrepancies as a static problem, DCPC can dynamically adapts to these variations. At its core, DCPC builds on the idea that semantically related inputs should yield similar label predictions, even in the presence of annotation biases. In this paper, we make the following key contributions:

- **Propose a self-supervised framework (DCPC) for addressing label preference inconsistencies**: DCPC dynamically adjusts prefix embeddings during fine-tuning to align semantically similar inputs, providing robustness against annotation biases.

- **Develop a preference-sensitive similarity mechanism, cross-layer prefix alignment, and a Preference Correction Module (PCM)**: These components detect and correct label discrepancies, ensuring consistent predictions across varying annotations.

- **Show superior performance over existing PEFT methods**: DCPC achieved state of the art(SOTA) results on multiple tasks and datasets, especially in handling subjective or biased annotations.

## 2 RELATED WORKS

**Parameter-Efficient Fine-Tuning**  As LLMs grow, fine-tuning becomes increasingly resource-intensive(Xin et al., 2024). Parameter-Efficient Fine-Tuning (PEFT) methods, like LoRA (Hu et al., 2021; Gao et al., 2024) and P-Tuning v2 (Liu et al., 2021), address this by updating only a small subset of parameters, while freezing the rest. Adapter-based methods (Houlsby et al., 2019; Chen et al., 2024) further reduce the parameter footprint by introducing bottleneck layers. While effective across benchmarks, these techniques do not address label preference inconsistencies across datasets.

**Prompt-Tuning Methods**  Prompt-tuning methods have emerged as a popular approach for adapting LLMs to various tasks without full model fine-tuning. These methods introduce learnable soft prompts that act as task-specific instructions, guiding the model during inference. Prefix-Tuning (Li & Liang, 2021; Liu et al., 2021; Vu et al., 2021; Ouyang et al., 2023) prepend trainable prefix embeddings to input sequences and internal layers, significantly improving model performance while reducing computational costs. Other variations modify internal components like attention mechanisms or bias terms (Tan et al., 2024). However, these techniques do not account for labeling inconsistencies across datasets. Our DCPC framework addresses this gap by dynamically adjusting prefix embeddings based on preference-sensitive similarity and cross-layer alignment, offering a more robust solution for handling heterogeneous datasets with varying annotation styles.

**Learning with inconsistent labels**  To the best of our knowledge, no existing work in the fine-tuning of LLMs has addressed the issue of inconsistent labels. In the context of traditional small- and medium-scale deep learning models, inconsistent labels have already posed a significant challenge for real-world applications (Rodrigues & Pereira, 2018; Chen et al., 2020). Several methods have been proposed, such as inferring the unknown true label of each instance from multiple noisy labels (Zhang et al., 2014). Majority Voting (MV) (Raykar & Yu, 2012) is a commonly used technique, which assumes that the labeling quality is balanced across the dataset—an assumption that is often unrealistic. Other approaches, such as RSVMI (Yang et al., 2023), LAWMV (Chen et al., 2022), and AALI (Zheng et al., 2021), utilize instance-specific features. However, these methods struggle when applied to LLM fine-tuning, where subjective or domain-specific annotations introduce more complex label inconsistencies, reflecting inherent biases or ambiguities rather than simple noise.

## 3 METHODS

### 3.1 PRELIMINARY

P-Tuning v2 is a parameter-efficient tuning technique designed for large pre-trained language models. Instead of fine-tuning the entire set of model parameters, P-Tuning v2 optimizes a small set of continuous task-specific prefix embeddings that are inserted into each layer of the transformer model. These prefix embeddings act as learnable prompts that guide the model in adapting to new downstream tasks while keeping the majority of the model parameters frozen.

For a given input sequence $x = \{x_1, x_2, \ldots, x_n\}$, the model first computes its embedding representation at each layer. Let $\mathbf{e}_x^l \in \mathbb{R}^d$ denote the embedding representation of $x$ at transformer layer $l$, where $d$ is the embedding dimension. In P-Tuning v2, a prefix embedding $\mathbf{P}_x^l \in \mathbb{R}^{m \times d}$ is learned, where $m$ represents the length of the prefix. These prefix embeddings are prepended to the input sequence embeddings before being fed into the transformer layers.

The modified input for layer $l$, combining both the prefix and the original token embeddings, is represented as:

$$\tilde{\mathbf{e}}_x^l = [\mathbf{P}_x^l; \mathbf{e}_x^l] \tag{1}$$

where $[\mathbf{P}_x^l; \mathbf{e}_x^l]$ denotes the concatenation of the prefix embedding $\mathbf{P}_x^l$ and the input embedding $\mathbf{e}_x^l$ along the sequence dimension.

The transformer layer processes this augmented input using the self-attention mechanism and feed-forward network, updating the hidden representations at each layer. The output of the transformer layer $l$, denoted as $\mathbf{h}_x^l$, is computed as:

$$\mathbf{h}_x^l = \text{TransformerLayer}^l([\mathbf{P}_x^l; \mathbf{e}_x^l]) \tag{2}$$

During training, the prefix embeddings $\mathbf{P}_x^l$ are learned for each layer $l$, while the pre-trained transformer model parameters are frozen.

Although P-Tuning v2 is highly efficient in task adaptation by optimizing the prefix embeddings, it does not explicitly address issues related to inconsistent label preferences across fine-tuning datasets. We conducted a toy experiment on the IMDB sentiment classification dataset, which contains movie reviews labeled with sentiments. The detailed design of this experiment can be found in A.2.

We analyze the layer-wise cosine similarity of the embeddings, the edit distance, and the KL-Divergence of the label preference distributions. The results are shown in Figure 2.

In the early layers (layers 1-5), the embeddings for $\mathbf{e}_A$ and $\mathbf{e}_B$ remain highly similar, as reflected in both the cosine similarity and the low edit distance. However, as the layers deepen, the predicted label preferences begin to diverge significantly. This suggests that, despite similar semantic representations, the model's predicted preferences are drifting, likely due to learned biases or inconsistencies in the training data.

This experiment reveals that semantically similar inputs can lead to inconsistent label preferences, which escalate as the model processes deeper layers. The increasing KL-Divergence suggests that the model's internal biases or preferences become more pronounced, even when the inputs remain similar in meaning. To counteract this effect, there is a need for a mechanism that

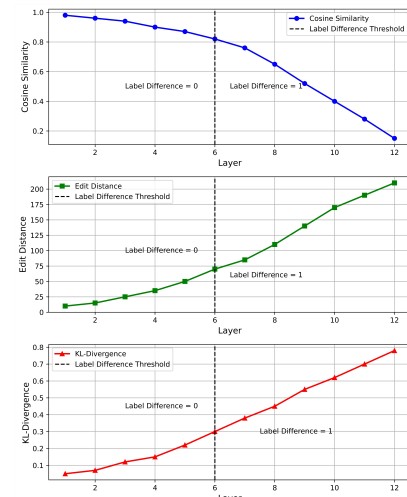

Figure 2: Toy experiment results analyzing the effects of label preference inconsistencies on semantically similar inputs.

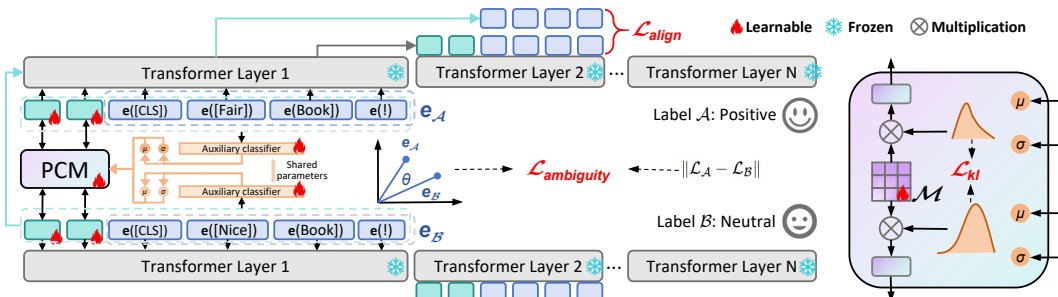

Figure 3: The overall pipeline of the proposed framework, Dynamic Cross-Layer Preference Correction (DCPC). The model first computes embeddings for two input sequences at each transformer layer. A preference-sensitive similarity mechanism compares these embeddings and triggers further steps if ambiguities arise. When label preference inconsistencies are detected, the cross-layer prefix alignment ensures that the embeddings are aligned across layers. Meanwhile, the Preference Correction Module (PCM) is activated to generate new prefix embeddings and correct for label preference discrepancies. The model learns to adapt its predictions dynamically by incorporating these corrections across layers.

realigns the model's predicted preferences with its embeddings, ensuring that the outputs remain consistent and less influenced by these learned biases.

## 3.2 OVERALL

The proposed framework, referred to as Dynamic Cross-Layer Preference Correction (DCPC) and illustrated in Figure 3, addresses label preference inconsistencies through a structured sequence of steps. For two input sequences $x_A$ and $x_B$, the model computes their respective embeddings $\mathbf{e}_A^l \in \mathbb{R}^d$ and $\mathbf{e}_B^l \in \mathbb{R}^d$ at transformer layer $l$, where $d$ is the embedding dimension.

The DCPC framework first evaluates the similarity between embeddings $\mathbf{e}_A^l$ and $\mathbf{e}_B^l$ using a preference-sensitive similarity mechanism. If $\cos(\theta)$ between $\mathbf{e}_A^l$ and $\mathbf{e}_B^l$ is high but $L_A \neq L_B$, it indicates a preference inconsistency, triggering corrective mechanisms.

Next, when significant discrepancies between labels $L_A$ and $L_B$ arise, the cross-layer prefix alignment mechanism is activated. This ensures alignment between both the prefix embeddings $\mathbf{P}_A^l$, $\mathbf{P}_B^l$, and token embeddings $\mathbf{T}_A^l$, $\mathbf{T}_B^l$ across all layers $l$. The goal is to maintain consistency between representations by minimizing differences in $\mathbf{P}_A^l$ and $\mathbf{P}_B^l$ through each transformer layer. In parallel, the Preference Correction Module (PCM) dynamically adjusts prefix embeddings $\mathbf{P}_{\text{new}}$ by leveraging a meta-matrix $\mathcal{M} \in \mathbb{R}^{m \times d}$, where $m$ is the prefix length. New prefixes are computed based on auxiliary classifier outputs and preference distributions $p(\mathbf{e}_A)$, $p(\mathbf{e}_B)$. The PCM minimizes discrepancies via adjustments to $\mathbf{P}_{\text{new}}$, ensuring alignment in the preference space.

This integration of cross-layer prefix alignment and preference correction dynamically corrects inconsistencies, making DCPC an efficient framework designed to optimize both inter-layer consistency and preference-driven discrepancies.

## 3.3 PREFERENCE-SENSITIVE SIMILARITY MECHANISM

To address the issue of inconsistent labeling preferences, we introduce a preference-sensitive similarity mechanism. This mechanism measures the representational similarity between input embeddings and compares it with the predicted labels to detect any discrepancies.

Given two input sequences, $x_A$ and $x_B$, their respective embeddings at layer $l$ are denoted as $\mathbf{e}_A^l \in \mathbb{R}^d$ and $\mathbf{e}_B^l \in \mathbb{R}^d$. To assess the similarity between these embeddings, we compute their cosine similarity:

$$\cos(\theta) = \frac{\mathbf{e}_A^l \cdot \mathbf{e}_B^l}{\|\mathbf{e}_A^l\|\|\mathbf{e}_B^l\|} \tag{3}$$

If the embeddings $\mathbf{e}_A^l$ and $\mathbf{e}_B^l$ are similar (i.e., $\cos(\theta) \geq \tau_{\cos}$ for a predefined threshold $\tau_{\cos}$), but their respective predicted labels $L_A$ and $L_B$ differ significantly, we use the label distributions (e.g., the softmax output probabilities) to measure the discrepancy between $L_A$ and $L_B$. Specifically, we compute the cross-entropy between the label distributions:

$$D_{\text{label}}(L_A, L_B) = -\sum_i L_A(i) \log L_B(i) \tag{4}$$

The ambiguity loss is defined as:

$$L_{\text{ambiguity}} = \mathbb{I}[\cos(\theta) \geq \tau_{\cos}] \cdot \cos(\theta) \cdot D_{\text{label}}(L_A, L_B) \tag{5}$$

Here, the indicator function $\mathbb{I}[\cos(\theta) \geq \tau_{\cos}]$ ensures that the ambiguity loss is only computed when the cosine similarity exceeds the threshold $\tau_{\cos}$. The term $\cos(\theta)$ acts as a weighting factor, amplifying the impact of label discrepancies when the embeddings are highly similar. This mechanism encourages the model to minimize label differences for semantically similar inputs during training.

### 3.4 CROSS-LAYER PREFIX ALIGNMENT

When the ambiguity loss $L_{\text{ambiguity}}$ exceeds a threshold $\tau_{\text{ambiguity}}$, indicating a substantial label preference misalignment, we activate the cross-layer prefix alignment mechanism. This step ensures that embeddings are aligned across transformer layers for inputs that exhibit high semantic similarity but have conflicting label predictions.

Let $\mathbf{P}_A^l$ and $\mathbf{T}_B^{l+1}$ represent the prefix embedding of input $A$ at layer $l$ and the token embedding of input $B$ at layer $l+1$, respectively. To align the representations, we concatenate the prefix of one input with the token embeddings of the other, forming new representations $\mathbf{C}_A^l$ and $\mathbf{C}_B^l$:

$$\mathbf{C}_A^l = \mathbf{P}_A^l \oplus \mathbf{T}_B^{l+1}, \quad \mathbf{C}_B^l = \mathbf{P}_B^l \oplus \mathbf{T}_A^{l+1} \tag{6}$$

To ensure alignment between these cross-layer representations, we define the alignment loss $L_{\text{align}}$ using the squared Euclidean distance:

$$L_{\text{align}} = \left\| \mathbf{C}_A^l - \mathbf{C}_B^l \right\| \tag{7}$$

Here, $\|\mathbf{C}_A^l - \mathbf{C}_B^l\|_2^2$ represents the squared Euclidean distance between the two concatenated representations. Minimizing this loss encourages the cross-layer embeddings $\mathbf{C}_A^l$ and $\mathbf{C}_B^l$ to be as close as possible in the embedding space, promoting consistency between semantically similar inputs.

**Theorem 1** (Asymptotic Consistency of Cross-Layer Prefix Alignment). *Assume two input sequences $x_A$ and $x_B$ have high semantic similarity at transformer layer $l$, i.e., their embeddings satisfy*

$$\cos(\theta(e_A^l, e_B^l)) \geq \tau_{cos}, \tag{8}$$

*where $e_A^l$ and $e_B^l$ are the embeddings at layer $l$. By applying the cross-layer prefix alignment mechanism, the prefix embeddings $P_A^l$ and $P_B^l$ will progressively converge in deeper layers $(l+1, l+2, \ldots)$, ensuring consistent predictions across the model's layers.*

The proof of Theorem 1 can be found in A.3.

### 3.5 PREFERENCE CORRECTION MODULE (PCM)

Simultaneously, when $L_{\text{ambiguity}}$ exceeds $\tau_{\text{ambiguity}}$, the Preference Correction Module (PCM) is activated to adjust the prefix embeddings and resolve label preference discrepancies. The PCM consists of two key components: auxiliary classifiers and a meta-matrix $\mathcal{M}$. The auxiliary classifiers are

responsible for predicting preference distributions for the input embeddings. The meta-matrix $\mathcal{M}$ stores learned patterns related to preference correction and is used to generate new prefix embeddings for each layer.

**Auxiliary Classifier Predictions** For each input embedding $\mathbf{e}_A$ and $\mathbf{e}_B$ with dimension $\mathbb{R}^d$, the auxiliary classifier predicts two parameters, $\mu \in \mathbb{R}^d$ (mean) and $\sigma \in \mathbb{R}^d$ (variance), which represent the distribution of preferences for each embedding. The shared parameters ensure consistency across transformer layers. These parameters are predicted as:

$$(\mu_A, \sigma_A) = \text{AuxClassifier}(\mathbf{e}_A), \quad (\mu_B, \sigma_B) = \text{AuxClassifier}(\mathbf{e}_B) \tag{9}$$

Here, $\mu_A \in \mathbb{R}^d$ and $\sigma_A \in \mathbb{R}^d$ are the predicted mean and variance for input $A$, while $\mu_B \in \mathbb{R}^d$ and $\sigma_B \in \mathbb{R}^d$ are for input $B$.

Once we have the predicted parameters, we sample a random noise vector $\epsilon \in \mathbb{R}^d$ from a standard normal distribution. Using these elements, we construct a preference distribution based on the predicted mean and variance. To ensure the stability and smoothness of this distribution, we apply a softmax operation:

$$p_{\text{pref}}(\mu, \epsilon) = \text{softmax}(\mu + \sigma \cdot \epsilon) \in \mathbb{R}^d \tag{10}$$

Here, $\sigma \cdot \epsilon \in \mathbb{R}^d$ introduces variability into the distribution by perturbing the predicted mean $\mu \in \mathbb{R}^d$.

**Generation of New Prefix Embeddings** The normalized preference distribution $p_{\text{pref}}(\mu, \epsilon) \in \mathbb{R}^d$ is then combined with the meta-matrix $\mathcal{M} \in \mathbb{R}^{m \times d}$, where $m$ represents the length of the new prefix embedding, and $d$ is the dimension of the embeddings. The new prefix embeddings $\mathbf{P}_{\text{new}} \in \mathbb{R}^m$ are generated via matrix multiplication:

$$\mathbf{P}_{\text{new}} = \mathcal{M} \cdot p_{\text{pref}}(\mu, \epsilon) \tag{11}$$

Here, $\mathcal{M} \in \mathbb{R}^{m \times d}$ is multiplied by $p_{\text{pref}}(\mu, \epsilon) \in \mathbb{R}^d$, resulting in a new prefix embedding $\mathbf{P}_{\text{new}} \in \mathbb{R}^m$. This operation adjusts the prefix embeddings based on the learned preference patterns encoded in $\mathcal{M}$ and the input embeddings' distributions.

**KL-Divergence Loss for Preference Alignment** To ensure that the newly generated prefix embeddings align with the original embeddings' preference distributions, we introduce a KL-divergence loss. This loss penalizes the divergence between the predicted preference distributions for $\mathbf{e}_A$ and $\mathbf{e}_B$:

$$L_{\text{KL}} = D_{\text{KL}}(p_A(\mu_A, \sigma_A) \| p_B(\mu_B, \sigma_B)) \tag{12}$$

The KL-divergence ensures that the distributions for the two inputs become closer, leading to more aligned prefix embeddings across layers.

Finally, The overall loss function is defined as:

$$L_{\text{total}} = \lambda_1 L_{\text{ambiguity}} + \lambda_2 L_{\text{align}} + \lambda_3 L_{\text{KL}} \tag{13}$$

where $\lambda_1$, $\lambda_2$, and $\lambda_3$ are hyperparameters controlling the relative importance of each loss term. The objective is to minimize label inconsistency while maintaining alignment across embedding layers and correcting for label preference discrepancies.

# 4 EXPERIMENTS

## 4.1 EXPERIMENTAL SETUP

**Datasets** We evaluate the performance of DCPC framework using a variety of datasets that involve subjective labeling or human preference discrepancies:(a) three tasks from SuperGLUE benchmark(**BoolQ**,**COPA**, and **ReCoRD**)(Wang et al., 2019). (b)two tasks from GLUE benchmark(**SST-2**

and **RTE**)(Wang, 2018). (c) **Alpaca** Dataset(Taori et al., 2023). For a detailed description of these datasets, see A.1.1.

Additionally, we extend these datasets with modified versions to introduce shifts in label preferences and biases, such as BoolQ-PreferenceShift(**BoolQ-PS**), COPA-BiasShift(**COPA-BS**),ReCoRD-Rephrase(**ReCoRD-R**), SST-2-PolarityShift( **SST-2-P**), RTE-EntailmentShift(**RTE-E**), and Alpaca-InstructionShift(**Alpaca-IS**). These variations allow us to simulate real-world annotator biases and inconsistencies. Detailed descriptions of the datasets and modifications can be found in the appendix (see A.1.1).

**Evaluation Metrics**   For **SST-2**, **RTE**, **BoolQ**, and **COPA**, we measure performance based on the accuracy of the model's predictions (denoted as **acc**), which reflects the proportion of correct answers compared to ground truth labels. For **ReCoRD**, we calculate both the F1 score and the exact match (EM) score. The final evaluation metric for ReCoRD is the average of these two scores (denoted as **f1-em**). For the **Alpaca** dataset and its modified versions, we leverage GPT-4o as an evaluator to assign a quantitative score to each response, based on coherence, completeness, and adherence to the task instructions. The average score provided by GPT-4o on a scale from 1 to 10 (denoted as **gpt-score**) is used as the primary performance metric for instruction-tuning tasks.

**Baselines**   We compare our Dynamic Cross-Layer Preference Correction (DCPC) with full-parameter fine-tuning (Full-FT) and several state-of-the-art PEFT methods. Representation modification methods include BitFit (Zaken et al., 2021), which adds trainable bias terms, and $(IA)^3$ (Liu et al., 2022a), which scales hidden representations using trainable vectors. Adapter-based methods, such as Houlsby-Adapter (Houlsby et al., 2019) and Learned-Adapter (Zhang et al., 2023b), add bottleneck layers for efficient tuning. Prompt-based tuning methods include P-Tuning v2 (Liu et al., 2021), LPT (Liu et al., 2022b), and PEDRO (Xie et al., 2024). We also evaluate LoRA (Hu et al., 2021) and its variant AdaLoRA (Zhang et al., 2023a), which use low-rank adaptation matrices with dynamic pruning. For a detailed overview of the baseline, please refer to A.1.2.

**Implementation Details**   All experiments are conducted using NVIDIA A100. For our main experiments, we fine-tune the LlaMA-2 models(Touvron et al., 2023), specifically the LlaMA-2 7B and LlaMA-2 13B models, as the backbone for the DCPC framework. We also conducted ablation experiments on Mistral-7B(Jiang et al., 2023). The predictions are generated using the standard language modeling (LM) head provided by the LlaMA-2 models. During inference, we apply beam search with a beam size of 3 to enhance the diversity and quality of generated outputs. The hyper-parameters of the DCPC framework are set as follows: (a) the length of the prefix embeddings $m$ is fixed at 16, (b) the meta-matrix $\mathcal{M}$ in the Preference Correction Module (PCM) is configured with dimensions $m \times d$, where $d = 4096$ for LlaMA-2 7B and $d = 5120$ for LlaMA-2 13B, corresponding to the hidden dimension of each model. (c) The cross-layer alignment similarity threshold $\tau_{cos}$ is set to 0.85, and the ambiguity loss threshold $\tau_{ambiguity}$ is set to 0.3.

We fine-tune the LlaMA-2 7B and 13B models using the HuggingFace Transformers library. The maximum sequence length is set to 2048 tokens for both models, and training runs for up to 10 epochs. The batch size is 16 for smaller datasets (e.g., SST-2 and RTE) and 64 for larger datasets (e.g., ReCoRD and BoolQ). We employ the AdamW optimizer with an initial learning rate of $1 \times 10^{-4}$, utilizing a linear learning rate decay and a warm-up phase covering 6% of the training steps. Evaluation is performed on the development set every 200 steps, and early stopping is applied if no improvement is observed after 10 evaluations. The best checkpoint based on the development set is used for final testing.

## 4.2 MAIN RESULTS

The experimental results on both the original and modified datasets are shown in Table 1 and Table 2, respectively.

**Performance on Original Datasets**   As shown in Table 1, our DCPC method consistently outperforms all baseline methods on the original datasets. Specifically, DCPC achieves the highest accuracy on **BoolQ** (88.9%), **COPA** (93.5%), **ReCoRD** (92.2%), **SST-2** (95.0%), and **RTE** (84.7%), demonstrating the effectiveness of our method in handling preference discrepancies in these tasks.

Table 1: Performance comparison of DCPC and baseline methods on original datasets. Results are median performance across five random seeds. The backbone is LlaMA-2 7B. Bold and underlined values represent the best and second-best results, respectively.

| Method | Tunable Params | BoolQ (acc) | COPA (acc) | ReCoRD (f1-em) | SST-2 (acc) | RTE (acc) | Alpaca (gpt-score) |
|---|---|---|---|---|---|---|---|
| Full-FT | 7B | 88.6 | 91.5 | 92.1 | 94.1 | 84.8 | 9.2 |
| P-Tuning v2 | 9.4M | 85.4 | 89.8 | 89.2 | 92.5 | 80.9 | 8.9 |
| LPT | 8.4M | 86.2 | 90.1 | 89.5 | 92.7 | 81.5 | 9.0 |
| Houlsby-Adapter | 9.5M | 86.5 | 90.3 | 89.7 | 92.9 | 81.8 | 9.1 |
| Learned-Adapter | 9.5M | 86.9 | 90.5 | 90.0 | 93.4 | 84.3 | 9.3 |
| LoRA | 10.0M | 86.7 | 90.8 | 90.2 | 93.5 | 82.3 | 9.2 |
| AdaLoRA | 10.0M | 87.1 | 91.0 | 91.8 | 93.6 | 82.7 | 9.2 |
| $(IA)^3$ | 9.8M | 86.6 | 90.6 | 90.1 | 93.2 | 82.0 | 9.4 |
| PEDRO | 8.9M | 88.1 | 92.3 | 91.7 | 94.7 | 84.2 | 9.3 |
| DCPC (ours) | 9.6M | **88.9** | **93.5** | **92.2** | **95.0** | **84.7** | **9.5** |

DCPC also achieves the best **gpt-score** of 9.5 on the Alpaca dataset, showing its superiority in instruction-following tasks. Among the baselines, the closest competitor is the PEDRO method, which also performs well but is consistently outperformed by DCPC across all datasets.

Table 2: Performance comparison of DCPC and baseline methods on modified datasets. Results are median performance across five random seeds. The backbone is LlaMA-2 7B. Bold and underlined values represent the best and second-best results, respectively.

| Method | Tunable Params | BoolQ-PS (acc) | COPA-BS (acc) | ReCoRD-R (f1-em) | SST-2-P (acc) | RTE-E (acc) | Alpaca-IS (gpt-score) |
|---|---|---|---|---|---|---|---|
| Full-FT | 7B | 82.4 | 88.5 | 88.4 | 90.1 | 80.7 | 8.7 |
| P-Tuning v2 | 9.4M | 78.0 | 86.1 | 85.9 | 87.5 | 77.9 | 8.4 |
| LPT | 8.4M | 78.5 | 86.4 | 86.2 | 87.8 | 78.3 | 8.5 |
| Houlsby-Adapter | 9.5M | 78.9 | 86.9 | 86.5 | 86.4 | 78.6 | 8.6 |
| Learned-Adapter | 9.5M | 79.2 | 86.8 | 87.1 | 88.3 | 78.9 | 8.7 |
| LoRA | 10.0M | 79.1 | 86.9 | 86.9 | 88.5 | 79.1 | 8.6 |
| AdaLoRA | 10.0M | 79.4 | 87.1 | 87.0 | 88.7 | 79.2 | 8.6 |
| $(IA)^3$ | 9.8M | 79.0 | 87.0 | 86.8 | 88.6 | 79.0 | 8.5 |
| PEDRO | 8.9M | 79.1 | 87.5 | 87.5 | 88.1 | 79.7 | 8.6 |
| DCPC (ours) | 9.6M | **86.1** | **91.7** | **91.9** | **92.8** | **83.7** | **9.4** |

**Performance on Modified Datasets**  Table 2 shows the performance of DCPC and baseline methods on the modified datasets, where preference shifts or biases have been introduced. DCPC again demonstrates its robustness, outperforming all baselines on the modified datasets as well. In particular, DCPC achieves the best performance on **BoolQ-PS** (86.1%), **COPA-BS** (91.7%), **ReCoRD-R** (91.9%), **SST-2-P** (92.8%), **RTE-E** (83.7%), and **Alpaca-IS** (9.4). The performance degradation of baseline methods on the modified datasets is more pronounced compared to DCPC, which shows a relatively smaller drop in performance. For example, Full-FT drops significantly from 88.6% to 82.4% on **BoolQ-PS**, whereas DCPC only drops from 88.9% to 86.1%. Similarly, on **COPA-BS**, Full-FT sees a large performance drop from 91.5% to 88.5%, while DCPC remains strong with a smaller drop to 91.7%. This highlights DCPC's ability to mitigate the impact of label preference shifts and biases effectively.

### 4.3 ABLATION STUDY

To assess the contribution of each component in the Dynamic Cross-Layer Preference Correction (DCPC) framework, we conduct an ablation study. We disable key components one at a time and evaluate the performance on both original and modified datasets. The following four ablated variants of DCPC are tested:

- **DCPC w/o CLPA**: The Cross-Layer Prefix Alignment (CLPA) is removed. This variant tests the model's ability to manage preference discrepancies without explicit cross-layer alignment.
- **DCPC w/o PCM**: The Preference Correction Module (PCM) is disabled. This assesses the impact of removing the module that corrects label preference discrepancies via prefix adjustments.
- **DCPC w/o Ambiguity Loss**: The ambiguity loss component is excluded, which measures how performance is impacted when the model does not explicitly minimize semantic similarity-based label discrepancies.
- **DCPC w/o CLPA & PCM**: Both CLPA and PCM are removed, leaving only the ambiguity loss mechanism in place. This acts as a minimal variant of DCPC, similar to a standard fine-tuning approach with ambiguity-aware adjustments.

We evaluate the ablation variants on modified datasets to determine the importance of each component in handling preference shifts. Table 3 presents the results.

Table 3: Ablation Study: Performance comparison of DCPC with different components disabled. Results are median performance across five random seeds. The backbone is LlaMA-2 7B. Bold and underlined values represent the best and second-best results, respectively. The values in parentheses represent the performance drop compared to the full DCPC model.

| Method | BoolQ-PS (acc) | COPA-BS (acc) | ReCoRD-R (f1-em) | SST-2-P (acc) | RTE-E (acc) | Alpaca-IS (gpt-score) |
|---|---|---|---|---|---|---|
| DCPC (Full) | **86.1** | **91.7** | **91.9** | **92.8** | **83.7** | **9.4** |
| DCPC w/o CLPA | 82.7 (-3.4) | 89.0 (-2.7) | 88.5 (-3.4) | 90.0 (-2.8) | 80.8 (-2.9) | 8.9 (-0.5) |
| DCPC w/o PCM | 81.2 (-4.9) | 88.5 (-3.2) | 87.0 (-4.9) | 89.5 (-3.3) | 79.1 (-4.6) | 8.9 (-0.5) |
| DCPC w/o Ambiguity Loss | 80.0 (-6.1) | 87.1 (-4.6) | 88.0 (-3.9) | 89.2 (-3.6) | 80.0 (-3.7) | 8.8 (-0.6) |
| DCPC w/o CLPA & PCM | 78.5 (-7.6) | 86.5 (-5.2) | 87.3 (-4.6) | 88.7 (-4.1) | 79.5 (-4.2) | 8.7 (-0.7) |

**Ablation study of DCPC**  The ablation study results in Table 3 highlight the critical contributions of each DCPC component. Removing Cross-Layer Prefix Alignment (CLPA) leads to a noticeable drop in performance, especially on ReCoRD-R (-3.4 f1-em) and BoolQ-PS (-3.4 acc), showing CLPA's importance in maintaining consistency across layers. The Preference Correction Module (PCM) is equally vital, with its removal causing a 4.9-point accuracy drop on BoolQ-PS and 4.6 points on RTE-E, underscoring its role in correcting preference discrepancies. Disabling ambiguity loss results in a sharper decline (e.g., -6.1 acc on BoolQ-PS), indicating its key role in reducing label inconsistencies. The largest performance decrease occurs when both CLPA and PCM are disabled, with a 7.6-point drop on BoolQ-PS and 5.2 points on COPA-BS, confirming the combined effectiveness of CLPA, PCM, and ambiguity loss.

**Ablation on the pretrained backbones**  We investigate the impact of different backbone models on the performance of the proposed DCPC framework. As shown in Table 4, the performance of DCPC remains robust across all backbone models, with

Table 4: Backbone model ablation study.

| Backbone Model | Params | BoolQ-PS (acc) | COPA-BS (acc) | ReCoRD-R (f1-em) |
|---|---|---|---|---|
| DCPC (LlaMA-2 7B) | 7B | 86.1 | 91.7 | **91.9** |
| DCPC (LlaMA-2 13B) | 13B | **86.4** | **92.0** | 91.7 |
| DCPC (Mistral-7B) | 7B | 85.7 | 91.8 | 91.8 |

LlaMA-2 13B achieving the highest overall accuracy in the BoolQ-PS and COPA-BS datasets.

## 4.4 ROBUSTNESS ANALYSIS

In this section, we analyze the robustness of the proposed DCPC framework by studying the impact of key hyperparameters on model performance. We focus on three primary hyperparameters: (1) the length of prefix embeddings ($m$), (2) the ambiguity loss threshold ($\tau_{ambiguity}$), and (3) the cross-layer prefix alignment similarity threshold ($\tau_{cos}$).

**Prefix Length** ($m$): The prefix length $m$ controls the dimensionality of the prefix embeddings inserted into each transformer layer. To study its effect, we vary $m$ from 8 to 24 and observe the changes in model performance across datasets. Figure 4 shows that as $m$ increases, the performance improves until saturation is reached at $m = 16$. Beyond this value, performance either stagnates or slightly declines, suggesting that overlong prefix embeddings may introduce noise and reduce the model's ability to capture meaningful preference shifts efficiently.

**Ambiguity Loss Threshold** ($\tau_{\text{ambiguity}}$): The ambiguity loss threshold $\tau_{\text{ambiguity}}$ determines when the Preference Correction Module (PCM) is triggered to correct label discrepancies. We experiment with $\tau_{\text{ambiguity}}$ values ranging from 0.1 to 0.5. As shown in Figure 4, a moderate value of $\tau_{\text{ambiguity}} = 0.3$ yields the best performance. Lower thresholds (e.g., $\tau_{\text{ambiguity}} = 0.1$) result in frequent activations of the PCM, potentially over-correcting minor discrepancies, while higher thresholds (e.g., $\tau_{\text{ambiguity}} = 0.5$) reduce the corrective impact of the PCM, leading to larger inconsistencies in the final predictions.

**Cosine Similarity Threshold** ($\tau_{\text{cos}}$): The cosine similarity threshold $\tau_{\text{cos}}$ is critical for determining when embeddings are considered semantically similar enough to trigger the ambiguity loss. We vary $\tau_{\text{cos}}$ from 0.7 to 0.95 to assess its impact on performance. Figure 4 shows that setting $\tau_{\text{cos}} = 0.85$ achieves optimal results. Lower values result in too many similarity comparisons being treated as high, leading to unnecessary corrective actions, while higher values decrease the number of corrective interventions, reducing the overall effectiveness of the framework.

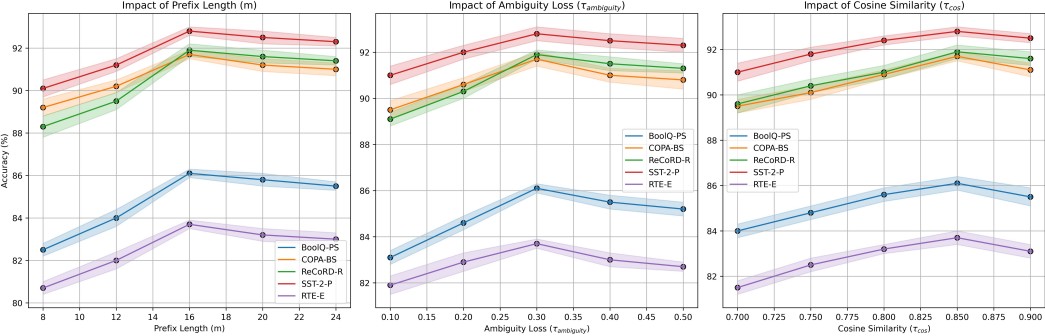

Figure 4: Impact of different hyperparameters on the performance of DCPC across multiple datasets. Subfigures show the effect of (a) prefix length ($m$), (b) ambiguity loss threshold ($\tau_{\text{ambiguity}}$), and (c) cosine similarity threshold ($\tau_{\text{cos}}$) on five datasets.

## 5 CONCLUSION AND FUTURE WORK

In this work, we proposed Dynamic Cross-Layer Preference Correction (DCPC) to address label preference inconsistencies in fine-tuning large language models. DCPC effectively reduces the impact of subjective labeling, outperforming state-of-the-art Parameter-Efficient Fine-Tuning (PEFT) methods across multiple datasets. It improves the alignment of semantically similar inputs, ensuring consistent label predictions, while highlighting the challenge of handling human preference shifts often overlooked in traditional fine-tuning techniques.

Our findings suggest future directions, including enhancing robustness against systematic biases in human annotations and exploring more efficient methods for aligning label preferences, such as innovations in prefix-tuning or preference correction mechanisms. Additionally, the ethical and practical implications of addressing label biases, particularly in sensitive domains, warrant further exploration.

Future work should extend DCPC's applicability to more diverse real-world datasets and refine corrective techniques, while also considering the societal and ethical challenges of mitigating subjective label biases in AI systems.

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

## A APPENDIX

### A.1 DETAILED EXPERIMENTAL SETUP

#### A.1.1 DESCRIPTION OF THE DATASETS

Original Datasets:

- **BoolQ (SuperGLUE)**: A yes/no question-answering task where answers are based on Wikipedia passages. Annotators may have subjective preferences when determining whether the passage supports a "yes" or "no" answer.
- **COPA (SuperGLUE)**: This task asks models to select the cause or effect of a given premise. Human judgment about cause-effect relationships is often subjective.
- ReCoRD (SuperGLUE): A reading comprehension task that involves identifying co-references in complex passages. Different annotators may interpret the text in unique ways, leading to inconsistent labels.
- **SST-2 (GLUE)**: A sentiment analysis task where sentences are labeled as positive or negative. Since sentiment labels are influenced by personal judgment, SST-2 is an ideal benchmark for testing how well DCPC manages subjective labeling.
- **RTE (GLUE)**: The Recognizing Textual Entailment (RTE) task asks whether one sentence entails another.
- **Alpaca Dataset**: This general-purpose instruction tuning dataset involves open-ended tasks where responses vary based on annotator preferences.

We extend the benchmark datasets with additional experimental setups to test the robustness of DCPC framework. In these additional setups, we introduce variations in label preferences by rephrasing or biasing the original annotations. The modified datasets allow us to simulate real-world conditions where annotator preferences and biases may influence labeling.

**BoolQ-PreferenceShift(BoolQ-PS)** For the BoolQ dataset, we use the GPT-3.5 API to rephrase the original yes/no labels into various styles, such as casual, formal, or expressive. The semantic meaning remains the same, but the phrasing of the answer is altered. The prompt used to generate the rephrased labels is as follows:

```
You are given a question and a yes/no answer.  Please
rewrite the answer in three different styles:  1)
Casual, 2) Formal, 3) Expressive.  Keep the meaning
of the answer the same.
Example:
Question:  "Is the sky blue?"
Answer:  "Yes."
Rephrased Answers:
1) Casual:  "Yeah, for sure."
2) Formal:  "Indeed, it is."
3) Expressive:  "Absolutely, without a doubt!"
```

**COPA-BiasShift(COPA-BS)** In the COPA dataset, we introduce an artificial bias in the selection of cause or effect by systematically shifting the chosen labels to favor human-related causes over natural causes. For each premise in the COPA dataset, the model must choose between two options:

one is the cause/effect related to human activity (e.g., "The person went to the store because..."), and the other is related to a natural event (e.g., "The rain caused flooding because..."). We introduce a bias $\beta$ that increases the likelihood of selecting human-related causes or effects.

Let the original probability of selecting cause/effect $o_i$ for a given premise be denoted as $P(o_i)$, where $i = 1$ represents the human-related option and $i = 2$ represents the natural-related option. The bias is introduced as a weighted probability shift, which is mathematically defined as follows:

$$P_{\text{biased}}(o_1) = \frac{P(o_1) + \alpha \cdot \mathbb{I}[o_1 \text{ is human-related}]}{P(o_1) + P(o_2) + \alpha} \tag{14}$$

$$P_{\text{biased}}(o_2) = \frac{P(o_2)}{P(o_1) + P(o_2) + \alpha} \tag{15}$$

where $P(o_1)$ and $P(o_2)$ represent the original, unbiased probabilities for the human-related and natural-related options, respectively. $\alpha$ is a bias factor that we introduce to shift preference toward human-related options. $\mathbb{I}[\cdot]$ is an indicator function that equals 1 when the condition inside it is true (i.e., when $o_1$ is a human-related option) and 0 otherwise. $P_{\text{biased}}(o_1)$ and $P_{\text{biased}}(o_2)$ represent the biased probabilities after applying the preference shift.

**ReCoRD-Rephrase(ReCoRD-R)**   For the ReCoRD dataset, we introduce variability in the expression of correct answers by using the GPT-3.5 API to generate alternative phrasings. While the core information and correctness of the answers remain unchanged, the phrasing and style are varied to simulate scenarios where different annotators might express the same answer in different ways. This tests how well the DCPC framework can reconcile these textual inconsistencies across layers. We use GPT-3.5 to rephrase the answers to the original questions in the ReCoRD dataset. Below is the prompt template used to generate the rephrased answers:

```
You are given a passage and a correct answer.  Please
rewrite the answer in three different ways while
keeping the meaning the same.  Try to express the
same information using different words and sentence
structures.
Example:
Passage:  "John went to the store to buy milk, but he
forgot to bring his wallet."
Answer:  "John forgot his wallet when he went to buy
milk."
Rephrased Answers:
1) "John went to the store for milk but didn't have
his wallet with him."
2) "When John went to purchase some milk, he realized
he had left his wallet behind."
3) "John didn't remember his wallet when he went to
buy milk."
```

The same prompt is applied to all answers in the dataset.

**SST-2-PolarityShift(SST-2-P)**   For sentiment analysis in the SST-2 dataset, we modify the sentiment labels by introducing slight shifts in their polarity. We adjust the labels of some positive reviews toward neutral sentiment, and negative reviews are softened to be less extreme. We define the sentiment labels for the SST-2 dataset as binary: $y_i \in \{0, 1\}$, where $y_i = 1$ represents a positive sentiment and $y_i = 0$ represents a negative sentiment. To introduce variability in the sentiment polarity, we apply a weighted shift to the original sentiment label $y_i$, producing a modified sentiment label $y_i'$.

For each sample, we introduce a shift parameter $\delta \in [0, 1]$ that represents the degree to which the sentiment label is altered. The modified sentiment label $y_i'$ is computed as:

$$y_i' = (1 - \delta) \cdot y_i + \delta \cdot \hat{y}_i \tag{16}$$

where $y_i$ is the original sentiment label (either 0 or 1). $\hat{y}_i$ is the opposite sentiment label of $y_i$ (i.e., $\hat{y}_i = 1 - y_i$). $\delta$ is a shift factor that controls the degree of sentiment modification. For example, $\delta = 0.2$ indicates a 20% shift toward the opposite sentiment.

To simulate a range of annotator subjectivity, we apply the polarity shift selectively to a portion of the dataset:

**Positive reviews** ($y_i = 1$): We shift some positive reviews toward neutral by decreasing the probability of a positive label using a lower $\delta$ value. For example, if $\delta = 0.3$, a positive review will be 30% closer to neutral, resulting in a softened sentiment of $y_i' = 0.7$.

$$y_i' = 0.7 \quad \text{(Shifted from fully positive to moderately positive)} \tag{17}$$

**Negative reviews** ($y_i = 0$): We soften some negative reviews by increasing the probability of a neutral sentiment. If $\delta = 0.4$, a negative review will be 40% softened, resulting in a less extreme sentiment label $y_i' = 0.4$.

$$y_i' = 0.4 \quad \text{(Shifted from fully negative to less negative)} \tag{18}$$

**RTE-EntailmentShift(RTE-E)** In the RTE dataset, we introduce biases into the entailment labels by systematically shifting the label distribution to prefer contradictions over entailments. The RTE dataset consists of premise-hypothesis pairs, where each pair is labeled as either Entailment ($y = 1$) or Contradiction/Neutral ($y = 0$). To introduce bias into the dataset, we adjust the labels of a subset of the pairs to favor contradictions. Specifically, we alter the probability distribution over the label space for each pair.

Let the original probability of the correct label for a given premise-hypothesis pair be denoted as $P(y_i)$, where $y_i = 1$ represents entailment and $y_i = 0$ represents contradiction or neutral. The biased probability $P_{\text{biased}}(y_i)$ is defined as:

$$P_{\text{biased}}(y_i = 0) = \frac{P(y_i = 0) + \beta \cdot \mathbb{I}[y_i = 1]}{P(y_i = 0) + P(y_i = 1) + \beta} \tag{19}$$

$$P_{\text{biased}}(y_i = 1) = \frac{P(y_i = 1)}{P(y_i = 0) + P(y_i = 1) + \beta} \tag{20}$$

where $P(y_i = 0)$ and $P(y_i = 1)$ are the original probabilities for the contradiction/neutral and entailment labels, respectively. $\beta$ is the bias factor that we introduce to increase the likelihood of selecting contradictions over entailments. $\mathbb{I}[\cdot]$ is an indicator function that equals 1 when the original label is entailment ($y_i = 1$) and 0 otherwise. $P_{\text{biased}}(y_i = 0)$ and $P_{\text{biased}}(y_i = 1)$ are the biased probabilities after applying the label preference shift.

This biasing process systematically shifts the probability distribution in favor of contradictions. For a subset of the dataset, we modify the labels based on the biased probabilities. For each premise-hypothesis pair, we select the final label $y_i'$ based on the biased distribution $P_{\text{biased}}(y_i)$:

$$y_i' = \begin{cases} 0, & \text{if } P_{\text{biased}}(y_i = 0) > P_{\text{biased}}(y_i = 1) \\ 1, & \text{otherwise} \end{cases} \tag{21}$$

**Alpaca-InstructionShift(Alpaca-IS):** For the Alpaca dataset, we introduce variability in the instructional outputs by using the GPT-3.5 API to generate responses in different styles, such as terse, elaborate, or conversational. While the core task remains unchanged, the stylistic variations in the instructions and responses introduce preference-driven differences.

To modify the instructional outputs and responses, we use GPT-3.5 to rephrase the original response in multiple styles. The following prompt template is designed to preserve the core task and meaning of the response while varying the style:

```
You are given an instruction and a response.  Please
rewrite the response in three different styles:  1)
Terse, 2) Elaborate, and 3) Conversational.  Keep the
meaning and the task the same, but vary the tone and
```

```
style of the response.
Example:
Instruction: "Write a summary of the novel '1984' by
George Orwell."
Response: "1984 is a dystopian novel about
totalitarianism."
Rephrased Responses:
1) Terse: "1984 is a dystopian story on totalitarian
rule."
2) Elaborate: "George Orwell's novel '1984' explores
a dystopian world under totalitarian rule, focusing on
themes of surveillance, freedom, and oppression."
3) Conversational: "So, 1984 is basically a story
where a totalitarian government controls everything,
and it's really all about how this impacts people's
lives."
```

### A.1.2 BASELINES

We compare our proposed Dynamic Cross-Layer Preference Correction (DCPC) framework with full-parameter fine-tuning (Full-FT) and several state-of-the-art Parameter-Efficient Fine-Tuning (PEFT) methods.

**Representation Modification Methods:** We include two common representation modification methods: (1) BitFit (Zaken et al., 2021), which introduces learnable parameters directly into the hidden representations by adding trainable bias terms; (2) $(IA)^3$ (Liu et al., 2022a), which modifies the hidden representations by scaling them using trainable vectors. Both methods keep the trainable vectors fixed across different samples. To adjust the number of tunable parameters, we initialize the vectors in a reduced dimension $r' < d_{\text{model}}$ and project them back to $d_{\text{model}}$ using a learnable matrix. For BitFit, $r' = 8$, and for $(IA)^3$, $r' = 16$.

**Adapter-Based Tuning:** We include two adapter-based methods as baselines: (1) Houlsby-Adapter (Houlsby et al., 2019), which is configured with a bottleneck dimension of 18, and (2) Learned-Adapter(Zhang et al., 2023b), which is configured with a bottleneck dimension of 36.

**Prompt-Based Tuning:** For prompt-based fine-tuning, we compare against: (1) P-Tuning v2(Liu et al., 2021), where the number of soft prompt tokens per layer is set to 64, (2) LPT (Liu et al., 2022b), which uses a bottleneck dimension of 1024 and a soft prompt length of 64 tokens, and (3) PEDRO(Xie et al., 2024) involves integrating a lightweight vector generator into each Transformer layer.

**LoRA and Its Variants:** We also consider LoRA (Hu et al., 2021) and its variant AdaLoRA(Zhang et al., 2023a) as baselines. For LoRA, the rank of the low-rank adaptation matrices is set to 4. For AdaLoRA, the initial rank is set to 8 per module, and half of the rank budget is dynamically pruned during fine-tuning.

### A.2 TOY EXPERIMENT: EXPLORING LABEL PREFERENCE INCONSISTENCIES IN SIMILAR INPUT EMBEDDINGS

The goal of this toy experiment is to investigate how semantically similar input sequences can lead to different label preference distributions under the P-Tuning v2 framework. We aim to explore whether prefix embeddings can effectively capture label preferences across similar inputs, and how inconsistencies arise.

### A.2.1 DATASET PREPARATION

**Dataset Selection** We use the IMDB sentiment classification dataset(Maas et al., 2011), where the sentiment labels (positive, neutral, negative) are often influenced by annotator preferences. This dataset is ideal for exploring the discrepancies in label preferences under P-Tuning v2.

**Sample Selection**   Two semantically similar review pairs are chosen:

- **Review A:** "The movie was enjoyable but not amazing." (Positive sentiment)
- **Review B:** "The film was okay, but nothing special." (Neutral sentiment)

These reviews have similar semantic meaning but are assigned different sentiment labels.

**Label Distributions**   We assume that for each input, the model generates a soft sentiment label distribution (e.g., probabilities of positive, neutral, and negative sentiment) instead of a hard label. These distributions represent the model's predicted preferences for each input sequence, which is influenced by the optimized prefix embeddings learned under P-Tuning v2.

### A.2.2   EXPERIMENT SETUP

**Layer-wise Embedding Calculation**   For each review, $x_A$ and $x_B$, we extract layer-wise embeddings $\mathbf{e}_A^l \in \mathbb{R}^d$ and $\mathbf{e}_B^l \in \mathbb{R}^d$ from a pre-trained transformer model (e.g., BERT), where $d = 768$ represents the embedding dimension. In P-Tuning v2, task-specific prefix embeddings are inserted into each transformer layer, and the embeddings $\mathbf{e}_A^l$ and $\mathbf{e}_B^l$ include the influence of these prefix embeddings.

**Label Preference Distribution**   At each layer $l$, the model with P-Tuning v2 computes the label preference distributions for both inputs using the softmax function over the model's output logits:

$$p(\mathbf{e}_A^l) = \text{Softmax}(f(\mathbf{e}_A^l)), \quad p(\mathbf{e}_B^l) = \text{Softmax}(f(\mathbf{e}_B^l)) \tag{22}$$

where $f(\mathbf{e}_A^l)$ and $f(\mathbf{e}_B^l)$ represent the logits for sentiment prediction at layer $l$. The resulting softmax outputs represent the predicted probability distributions over sentiment categories, which reflect how well the task-specific prefix embeddings capture label preferences.

**KL-Divergence Calculation**   We measure the divergence between the predicted label distributions for the two inputs at each layer using KL-Divergence:

$$L_{\text{KL}}(p_A, p_B) = D_{\text{KL}}(p(\mathbf{e}_A^l) \| p(\mathbf{e}_B^l)) \tag{23}$$

This quantifies how much the predicted label distributions for the two inputs deviate, even though their embeddings remain similar. The goal is to assess how much P-Tuning v2's prefix embeddings contribute to such discrepancies in label preferences.

### A.2.3   QUANTITATIVE ANALYSIS: LAYER-WISE EMBEDDING AND PREFERENCE DISTRIBUTION CHANGES

We analyze the layer-wise cosine similarity of the embeddings, the edit distance, and the KL-Divergence of the label preference distributions. The results are summarized in Table 5, demonstrating how prefix embeddings under P-Tuning v2 influence the emergence of preference inconsistencies.

### A.3   PROOF OF THEOREM 1

*Proof.* We aim to prove that, under the cross-layer prefix alignment mechanism, the difference between the prefix embeddings $P_A^{l+k}$ and $P_B^{l+k}$ decreases as we move to deeper layers $l + k$, where $k \geq 1$. Specifically, we will show that:

$$\|P_A^{l+k} - P_B^{l+k}\|_2 \leq \rho^k \|P_A^l - P_B^l\|_2, \tag{24}$$

for some constant $\rho \in [0, 1)$, indicating exponential decay of the difference, leading to convergence.

Let $e_A^l, e_B^l \in \mathbb{R}^d$ be the embeddings of inputs $x_A$ and $x_B$ at layer $l$. Let $P_A^l, P_B^l \in \mathbb{R}^{m \times d}$ be their respective prefix embeddings at layer $l$, where $m$ is the prefix length.

The cross-layer prefix alignment mechanism defines the concatenated embeddings:

$$C_A^l = P_A^l \oplus T_B^{l+1}, \quad C_B^l = P_B^l \oplus T_A^{l+1}, \tag{25}$$

Table 5: Layer-wise Cosine Similarity, Edit Distance, KL-Divergence, and Label Prediction Differences

| Layer $l$ | Cosine Similarity | Edit Distance | KL-Divergence | Label Difference (Prediction) |
|---|---|---|---|---|
| 1 | 0.98 | 10 | 0.05 | 0 |
| 2 | 0.96 | 15 | 0.07 | 0 |
| 3 | 0.94 | 25 | 0.12 | 0 |
| 4 | 0.90 | 35 | 0.15 | 0 |
| 5 | 0.87 | 50 | 0.22 | 0 |
| 6 | 0.82 | 70 | 0.30 | 1 |
| 7 | 0.76 | 85 | 0.38 | 1 |
| 8 | 0.65 | 110 | 0.45 | 1 |
| 9 | 0.52 | 140 | 0.55 | 1 |
| 10 | 0.40 | 170 | 0.62 | 1 |
| 11 | 0.28 | 190 | 0.70 | 1 |
| 12 | 0.15 | 210 | 0.78 | 1 |

where $\oplus$ denotes concatenation, and $T_A^{l+1}, T_B^{l+1}$ are the token embeddings at layer $l+1$.

The alignment loss at layer $l$ is:

$$L_{\text{align}}^l = \|C_A^l - C_B^l\|_2^2. \tag{26}$$

The training objective includes minimizing $L_{\text{align}}^l$ to encourage $C_A^l$ and $C_B^l$ to become closer.

This minimization updates $P_A^l$ and $P_B^l$ such that:

$$P_A^l \leftarrow P_A^l - \eta \frac{\partial L_{\text{align}}^l}{\partial P_A^l}, \quad P_B^l \leftarrow P_B^l - \eta \frac{\partial L_{\text{align}}^l}{\partial P_B^l}, \tag{27}$$

where $\eta$ is the learning rate.

The transformer layer updates the embeddings using a function $f$, which includes attention and feed-forward networks:

$$P_A^{l+1} = f(P_A^l), \quad P_B^{l+1} = f(P_B^l). \tag{28}$$

Our goal is to analyze $\|P_A^{l+1} - P_B^{l+1}\|_2$.

Assume the function $f$ is Lipschitz continuous with Lipschitz constant $L_f > 0$:

$$\|f(u) - f(v)\|_2 \le L_f \|u - v\|_2, \quad \forall u, v \in \mathbb{R}^{m \times d}. \tag{29}$$

Using the Lipschitz property:

$$\|P_A^{l+1} - P_B^{l+1}\|_2 = \|f(P_A^l) - f(P_B^l)\|_2 \le L_f \|P_A^l - P_B^l\|_2. \tag{30}$$

Applying the same reasoning recursively for layers $l+1, l+2, \ldots, l+k$:

$$\|P_A^{l+k} - P_B^{l+k}\|_2 \le L_f^k \|P_A^l - P_B^l\|_2. \tag{31}$$

Since $L_f > 0$, the difference decreases exponentially if $L_f < 1$.

The high semantic similarity of $x_A$ and $x_B$ at layer $l$ implies $\|e_A^l - e_B^l\|_2$ is small. Therefore, the initial difference $\|P_A^l - P_B^l\|_2$ is small due to the alignment loss minimization.

If $L_f < 1$, then as $k \to \infty$:

$$\|P_A^{l+k} - P_B^{l+k}\|_2 \to 0. \tag{32}$$

This indicates that $P_A^{l+k}$ and $P_B^{l+k}$ converge. $\qquad \square$

As the prefix embeddings converge, the model's outputs (predictions) based on these embeddings also become consistent. The transformation function $f$ propagates the alignment through the network, reducing discrepancies.

