# OpenReview forum: "Dynamic Cross-Layer Prefix Alignment for Resolving Label Preference Discrepancies in LLMs Fine-Tuning"
_ICLR.cc/2025/Conference — Submitted to ICLR 2025_

### Official Review · Reviewer_FZmj · 2024-10-17

**Soundness:** 3
**Presentation:** 3
**Contribution:** 2
**Rating:** 5
**Confidence:** 3

**Summary:**

This paper introduces the Dynamic Cross-Layer Preference Correction (DCPC) framework. This framework is designed to address the issue of inconsistent label preferences during fine-tuning of large language models (LLMs), particularly when annotations come from various sources with differing styles or biases.
The paper demonstrates that DCPC outperforms state-of-the-art parameter-efficient fine-tuning (PEFT) methods, especially in handling datasets with subjective or biased annotations.

**Strengths:**

1. The paper addresses a highly specific but underexplored problem—label preference discrepancies in fine-tuning large language models (LLMs)—which has a substantial impact on the robustness and generalization of models in real-world applications.

2. The introduction of Dynamic Cross-Layer Preference Correction (DCPC) is novel.  Extensive experiments across multiple datasets, including both original and modified versions, demonstrate the effectiveness of  DCPC.

3. The authors meticulously conduct an ablation study, isolating the contributions of different components (e.g., cross-layer prefix alignment, preference correction module), which strengthens the validation of their method.

**Weaknesses:**

1. The paper does not include comparisons with standard "Learning with inconsistent labels" methods like Majority Voting (MV). Without these comparisons, it is difficult to fully assess the performance advantage of DCPC.

2. The paper introduces several components, such as the cross-layer prefix alignment and Preference Correction Module (PCM), but lacks in-depth theoretical or empirical justification for these choices.

**Questions:**

The questions that I consider are mentioned in the above weakness section.

---

### Official Review · Reviewer_Ggxq · 2024-11-02

**Soundness:** 2
**Presentation:** 3
**Contribution:** 3
**Rating:** 5
**Confidence:** 4

**Summary:**

The paper presents the Dynamic Cross-Layer Preference Correction (DCPC) framework designed to mitigate label preference discrepancies in fine-tuning large language models (LLMs). It addresses the critical issue of inconsistent labeling among annotators, which can hinder model robustness and generalization.

**Strengths:**

1. Innovative Framework: The DCPC framework introduces novel mechanisms like cross-layer prefix alignment and a Preference Correction Module to effectively address label preference discrepancies, enhancing model robustness.
2. Comprehensive Evaluation: The paper provides extensive experiments across various tasks and datasets, demonstrating the effectiveness and robustness of DCPC in handling label preference shifts, supported by insightful ablation studies.

**Weaknesses:**

1. The sample size of the author's toy experiment is noticeably small. It is recommended to provide brief examples in the main text while adding some sample experiments in the appendix to enhance persuasiveness. The content of the experiments is also confusing; the calculations performed by the authors seem to have little direct relationship with P-Tuning V2. The conclusions drawn from the results in Figure 2 are easy to understand, but this soft distribution cannot directly prove this conclusion. As the layer depth increases, the cosine similarity of most sentences will decrease, and this should be contrasted.
2. Relying on generative models (such as GPT-3.5) to simulate label preference changes may introduce potential biases or inconsistencies that could affect the authenticity of the evaluation. Although the paper compares various baseline methods, the limited selection of large language model backbones involved might introduce some bias. Including more large language models as backbones could provide a more comprehensive demonstration of DCPC’s advantages.
3. Analysis Issue in Figure 2 (Section 3.1): In analyzing Figure 2, the authors conclude that even semantically similar inputs may lead to significantly different predictions due to low cosine similarity  in the later layers. They attribute this to learning bias or dataset inconsistency. However, I believe this phenomenon is a characteristic of large models themselves; even with semantically similar inputs, token differences in the later layers are often substantial. Moreover, the paper does not clarify whether this method can address this specific issue. The authors have not ruled out the possibility that, even without dataset inconsistencies and with semantically similar inputs, the difference between eA and eB in the later layers could still be large.
4. Handling of Multi-bench Scenarios: The methods section describes the process for handling two input sequences, but it does not clearly explain how the approach would work for more than two input sequences.
5. Execution Strategy of the Algorithm: The method introduced in this paper can only handle two input sequences at a time. If there is no label preference issue between each pair of input sequences, might the effectiveness of this method decrease? Would the method perform better if semantically similar inputs were paired in advance? It would be helpful if the paper addressed these questions.

**Questions:**

1. What role does cross-layer prefix alignment play in this? What is the significance of embedding convergence? Why is this component absent?
2. The authors' experimental results demonstrate the advantages of their method across various tasks, but how does this prove that DCPC effectively mitigates the impact of label preference changes and biases?

---

### Official Review · Reviewer_pkpN · 2024-11-03

**Soundness:** 2
**Presentation:** 3
**Contribution:** 2
**Rating:** 3
**Confidence:** 3

**Summary:**

This paper focuses on efficient fine-tuning and introduces a dynamic cross-layer preference correction method DCPC. Specifically, the DCPC method consists of a preference-sensitive similarity mechanism, cross-layer prefix alignment, and a preference correction module. The experimental results across multiple tasks show that the DCPC method outperforms some PEFT methods in handling label preference discrepancies.

**Strengths:**

1. The paper is well-organized and easy to read with clear explanations of the technical details.
2. The idea of handling label preference discrepancies is intuitive and the proposed method DCPC is complete and shows effectiveness in the experimental results.

**Weaknesses:**

1. The authors analyze the layer-wise cosine similarity of the embeddings between two semantic similar samples to verify the label preference discrepancies. However, there is a lack of experimental verification or theoretical analysis to illustrate the relationship between embedding vector similarity and semantic relevance. I believe this is necessary because it is the foundation of the follow-up proposed approach.
2. The universality of the proposed method is limited, as it only supports prefix-learning methods like P-tuning v2. In addition , this method necessitates the use of two semantically similar examples for training, making the training process costly.
3. The experimental results are not entirely convincing, as most of the evaluation tasks are classification tasks, and experiments are only conducted on llama models. It would be beneficial to verify the method's effectiveness on more evaluation tasks, such as code and mathematical reasoning tasks, and a broader range of model types and sizes.

**Questions:**

See Weaknesses.

---

### Official Review · Reviewer_qpEH · 2024-11-04

**Soundness:** 2
**Presentation:** 3
**Contribution:** 2
**Rating:** 5
**Confidence:** 3

**Summary:**

To address the issue of label preference discrepancies in fine-tuning language models, the paper proposes a method named Dynamic Cross-Layer Preference Correction (DCPC), which is a parameter-efficient fine-tuning approach and a self-supervised learning framework.

DCPC consists of a preference-sensitive similarity mechanism, cross-layer prefix alignment, and a Preference Correction Module.

Evaluated on different datasets, DCPC can outperform the previous parameter-efficient tuning baselines and perform well in dealing with label discrepancies.

**Strengths:**

Testing across different datasets, the proposed method, DCPC, demonstrates superior performance in parameter-efficient tuning and excels in addressing label discrepancies.

**Weaknesses:**

1. **Insufficient Explanation of Label Preference Discrepancy vs. Noisy Label Learning**: The paper lacks a clear explanation of how label preference discrepancy differs from noisy label learning. While label discrepancy can be viewed as a type of label noise, the paper does not address or clarify the similarities and differences between the two.


2. **Missing Baselines for Noisy Label Learning**: The baseline comparisons do not include methods specifically developed for learning from noisy labels, such as the co-teaching approach (https://arxiv.org/abs/1804.06872). The proposed methods should be evaluated against these noisy label learning baselines to provide a more comprehensive comparison.

**Questions:**

How does the label discrepancy occur in practice?

---

### Meta-Review · Area_Chair_kwjf · 2024-12-06

**Metareview:**

Overall reviewers agree that DCPC presents a novel approach addressing label preference discrepancies, with promising results, although there are areas for improvement in explanations and comparisons. In particular,

- The paper lacks a clear explanation of how label preference discrepancy differs from noisy label learning and does not include comparisons with noisy label learning baselines.
- There is insufficient experimental verification or theoretical analysis of the relationship between embedding vector similarity and semantic relevance.
- The universality of DCPC is limited, supporting only prefix-learning methods and requiring semantically similar examples for training, which can be costly.
- The sample size in toy experiments is small, and the reliance on generative models may introduce biases.
- The paper does not include comparisons with standard "Learning with inconsistent labels" methods like Majority Voting (MV), and lacks in-depth justification for the introduced components.

**Additional Comments On Reviewer Discussion:**

None

---

### Decision · Program_Chairs · 2025-01-22

Reject